# Airway Epithelial Dysfunction in Asthma: Relevant to Epidermal Growth Factor Receptors and Airway Epithelial Cells

**DOI:** 10.3390/jcm9113698

**Published:** 2020-11-18

**Authors:** Hideki Inoue, Kaho Akimoto, Tetsuya Homma, Akihiko Tanaka, Hironori Sagara

**Affiliations:** Division of Respiratory Medicine and Allergology, Department of Medicine, Showa University School of Medicine, Hatanodai, Shinagawa-ku, Tokyo 142-8666, Japan; k_akimoto@med.showa-u.ac.jp (K.A.); tetsuya.homma@med.showa-u.ac.jp (T.H.); tanakaa@med.showa-u.ac.jp (A.T.); sagarah@med.showa-u.ac.jp (H.S.)

**Keywords:** asthma, airway epithelial cells, epidermal growth factor receptor, ErbB2, ErbB3, ErbB4

## Abstract

Airway epithelium plays an important role as the first barrier from external pathogens, including bacteria, viruses, chemical substances, and allergic components. Airway epithelial cells also have pivotal roles as immunological coordinators of defense mechanisms to transfer signals to immunologic cells to eliminate external pathogens from airways. Impaired airway epithelium allows the pathogens to remain in the airway epithelium, which induces aberrant immunological reactions. Dysregulated functions of asthmatic airway epithelium have been reported in terms of impaired wound repair, fragile tight junctions, and excessive proliferation, leading to airway remodeling, which contributes to aberrant airway responses caused by external pathogens. To maintain airway epithelium integrity, a family of epidermal growth factor receptors (EGFR) have pivotal roles in mechanisms of cell growth, proliferation, and differentiation. There are extensive studies focusing on the relation between EGFR and asthma pathophysiology, which describe airway remodeling, airway hypermucus secretion, as well as immunological responses of airway inflammation. Furthermore, the second EGFR family member, erythroblastosis oncogene B2 (ErbB2), has been recognized to be involved with impaired wound recovery and epithelial differentiation in asthmatic airway epithelium. In this review, the roles of the EGFR family in asthmatic airway epithelium are focused on to elucidate the pathogenesis of airway epithelial dysfunction in asthma.

## 1. Introduction

Asthma is a chronic airway disease characterized as pathological features of chronic airway inflammation, airway reversibility, and airway remodeling [1,2,3,4]. Airway epithelium plays central roles in asthma pathogenesis as the first barrier from external environmental pathogens, including allergens, chemical mediators, cigarette smoke, bacteria, and viruses to protect host homeostasis [5]. To repair airway epithelium disrupted by external stimuli, various growth factors, including epidermal growth factor receptors (EGFRs), keratinocyte growth factor (KGF) [6], and human hepatocyte growth factor (hHGF) [7] are expressed in airway epithelium. EGFR family is indispensable, especially in asthmatic airways, which are vulnerable against inhaled substances [8]. In asthmatic airway epithelium, expression pattern of the EGFR family is different from healthy airway epithelium, and the impaired expression of EGFRs is considered relevant to asthma pathogenesis in terms of airway inflammation, airway remodeling, airway hyperresponsiveness (AHR), and airway mucus secretion. In this review, we focus on the relationship between asthma pathophysiology and airway epithelial EGFRs as potentially novel therapeutic targets for asthma.

## 2. Disrupted Airway Function of Asthma Epithelium

Airway epithelium is the first defense wall between the external air and the internal body. It plays a role as not only a physical barrier, but also as a signal sensor to coordinate immunological responses of signal transduction for excluding pathogens, including bacteria, viruses, fungi, and allergens [5]. Some of those pathogens damage airway epithelial cells and invade airway tissues. Those external stimuli elicit migration of inflammatory immune cells, including dendritic cells, lymphocytes, mast cells and neutrophils, as well as eosinophils whose inflammatory mediators induce local inflammation of the airways to eliminate and deactivate external pathogens. Once airway epithelium is disrupted by physical damage, inflammatory mediators from inflammatory immune cells, or proteases from external pathogens, the repair process for damaged airway epithelial cells is initiated by interactions between the immune cells and the airway epithelium, or epithelial mesenchymal tropic units, to maintain epithelial integrity [9]. 

In asthmatic airway epithelium, two factors could be involved with the disrupted airway integrity: (1) dysfunctional repair processes, and (2) vulnerability to external stimuli. First, “dysfunctional repair process” was observed in asthmatic airway epithelial cells. The repair process of the tissue is comprised of cell proliferation, cell migration, suppression of apoptosis, and cell differentiation. Each of those processes is indispensable to achieve airway epithelial repair, and delayed airway repair caused by disrupted repair processes allows external pathogens access to remaining airway tissue, which results in prolonged airway inflammation. In in vitro wound healing models using asthmatic airway epithelial cells, impaired cell proliferation and migration after mechanical wounding was observed and this finding could be a factor for severe asthma phenotype [10,11]. Secondly, “vulnerability to external stimuli” was also observed in asthmatic airway epithelium as shown in impaired tight junction formation [8]. Dysfunctional airway epithelium in asthmatics could allow easier access of external pathogens to airway submucosa and this further leads to chronic airway inflammation involved with asthma pathogenesis.

## 3. Roles of Epidermal Growth Factor Receptor Family in Asthma

In the process of airway epithelial cell proliferation and differentiation, a family of EGFR is evolutionally conserved and plays an important role. EGFRs are also involved in maintaining airway homeostasis and wound healing processes. EGFRs are comprised of four subtypes including EGFR, ErbB2 (erythroblastosis oncogene B2, also known as HER2, neu, or CD340), ErbB3 (HER3), and ErbB4 (HER4). EGFR is a glycoprotein with typical transmembrane tyrosine kinase receptors and is activated by its autophosphorylation with its ligand bindings, and subsequent homo- or hetero-oligomerization with other EFGR receptors. There are multiple ligands for each EGFR; for example, epidermal growth factor (EGF), heparin-binding EGF, amphiregulin, epiregulin, neuregulin (heregulin), epiregulin and transforming growth factor (TGF)-α [12]. Phosphorylated kinase domains of EGFRs allosterically activate a counter partner of EGFR receptors [13]. The receptor activation initiates downstream signaling cascades, which regulate cell proliferation, growth, differentiation, and survival by transducing signals through pathways including the Ras/Raf/mitogen-activated protein kinase (MAPK) pathway, the phosphatidylinositol 3-kinase (PI3K)/Akt pathway, and the phospholipase C (PLC)/protein kinase C (PKC) pathway [14]. There is extensive research about EGFRs in the context of cancers, but EGFRs are also essential for maintaining airway epithelial integrity and are involved in asthma pathogenesis (Table 1).

### 3.1. EGFR

#### 3.1.1. Upregulation of EGFR in Asthmatic Airway Epithelium

In epithelial repair processes, EGFR plays central roles by contributing cell proliferation, growth, and differentiation to maintain airway epithelial integrity. In asthmatic airway epithelium, EGFR expression increases upon damaging airway epithelium due to sustained airway inflammation. In ex vivo analysis, EGFR and EGF expression were increased in airway epithelium in autopsied or surgically resected lungs from asthmatic patients compared to those from control subjects [37]. Puddicombe et al. investigated the relationship between EGFR expression and airway epithelial wounding [15]. They reported that EGFR expression in airway epithelial cells increased in asthma, more prominent in severe asthmatic airway epithelium. In an in vitro study, phosphorylation of EGFR increased in scrape-wounded monolayer airway epithelial cells. EGFR-selective inhibitor of AG1478 inhibited wound repair on scraped monolayer epithelial cells with upregulated TGF-β production. The ligands of EGFR including amphiregulin and heparin-binding EGF (HB-EGF) were expressed more in airway epithelium among asthmatics than in the control group [21]. In in vitro experiments using airway epithelial cell-line (BEAS-2B), normal human bronchial epithelial cells (NHBE), and primary human airway epithelial cells, histamine stimulated upregulation and releasing of amphiregulin and HB-EGF, and histamine also increased EGFR phosphorylation and goblet cell differentiation. Additionally, amphiregulin and HB-EGF, which was stimulated by histamine, induced proliferation and migration of airway smooth muscle cells. The release of histamine would contribute to airway remodeling by subsequent excretion of EGFR ligands, such as amphiregulin and HB-EGF followed by EGFR phosphorylation [21]. 

#### 3.1.2. EGFR and Asthma in Animal Model

There is extensive research using an in vivo allergic animal model regarding airway epithelial EGFR expression and allergic airway inflammation. Ovalbumin (OVA)-induced allergic mice showed eosinophilia and increased T helper type 2 (Th2) cytokines including interleukin (IL)-4 and IL-13 in bronchoalveolar lavage fluid with airway hyperresponsiveness (AHR) [16]. Treatment with the EGFR inhibitor, gefitinib, resulted in improvement of eosinophilic airway inflammation and AHR with reduced expression of EGFR and its downstream pathway of PI3K/Akt [16]. In a in vivo experiment using the house dust mite (HDM)-induced allergic asthma mice model, expression of EGFR increased after HDM inhalation, which was inhibited by treatment with EGFR inhibitor, erlotinib [18]. HDM inhalation for 6 weeks caused AHR and airway remodeling assessed by airway smooth muscle thickening and treatment with erlotinib ameliorated AHR and airway remodeling. In another study, using the OVA-induced rat asthma model reported that ovalbumin sensitization and challenge resulted in airway remodeling, epithelial cell proliferation, and goblet cell hyperplasia with increased HB-EGF secretion, but not EGFR expression in the airway epithelium [22]. Intratracheal administration of leukotriene (LT) D4 reproduced the effects of airway remodeling as OVA induction. The treatment with EGFR inhibitor (AG1478) improved airway remodeling, and epithelial and goblet cell proliferation induced by OVA challenge or LTD4 administration, which suggested that EGFR expression is the downstream effect of cysteinyl leukotriene-1 receptor-EGFR axis. These results indicate that EGFR expression is involved with allergic airway inflammation, AHR, and airway remodeling induced by OVA or HDM and blockade of EGFR pathway may ameliorate allergic airway inflammation.

#### 3.1.3. EGFR and Airway Mucus Production

EGFR expression on airway epithelial cells also involves airway mucus production, which is a clinical feature of chronic asthma shown as airway hypermucus secretion by goblet cell hyperplasia. In vitro activation of EGFR by its ligands including EGF, TNF-α, and TGF-α increased MUC5AC gene and protein expression in airway epithelial cells [25]. The treatment by an EGFR inhibitor ameliorated MUC5AC expression induced by these EGFR ligands. These results were also observed using the in vivo asthma model of OVA-sensitized rats [25]. EGFR is also activated by bacterial components of lipopolysaccharide (LPS), which is often used as bacterial-induced asthma exacerbation model. The other in vitro study showed that stimulation by LPS or TNF-α to human bronchial epithelial cells (NCI-H292 cell) induced MUC5AC and IL-8 gene and protein expression. This LPS-induced upregulation of MUC5AC and IL-8 was inhibited by AG1478 treatment [26]. Another study also showed that LPS-induced EGFR activation induced MUC5AC gene and protein expression in airway epithelial cells (16HBE) [30]. EGFR inhibitor (AG1478) decreased LPS-induced MUC5AC expression. In in vivo experiments using rat chronic rhinosinusitis (CRS) models made by intranasal inhalation of LPS, intraperitoneal or intranasal treatment with AG1478 improved goblet hyperplasia and mucus hypersecretion in airway tissue [26]. EGFR-related epithelial barrier dysfunction would involve airway mucus production. Recent studies showed that EGFR activation in vitro and in vivo reduced claudin1 expression and increased MUC5AC expression with airway inflammation and airway hyperreactivity [38]. The EGFR receptor inhibitor, erlotinib, recovered airway inflammation and claudin1 expression. Aberrant expression of EGFR would disrupt airway epithelial barrier function and result in airway hypermucus secretion. In clinical aspects, a positive correlation between EGFR and MUC5AC expression was confirmed using bronchial mucosal biopsy specimens from human subjects divided into a healthy control group and a group with asthma [27]. The relationship between EGFR activation and mucus secretion was also confirmed using sputum samples from acute asthma attack of childhood asthma [29]. EGF and amphiregulin levels increased in sputum during acute childhood asthma compared to those participants with a stable condition. EGF stimulated the proliferation of normal human bronchial epithelial cells (NHBE) as well as bronchial smooth muscle cells and lung fibroblasts, but amphiregulin induced cell proliferation only in NHBE. Co-stimulation of EGF and amphiregulin in NHBE resulted in goblet cell metaplasia, shown as MUC5AC protein expression. These results indicate that overexpression of EGFR results in airway remodeling and hypermucus secretion with excessive airway inflammation. IL-13 is also known as a major cytokine involved in mucus overproduction in asthmatic airways. A study using a mouse model of chronic viral infection showed that persistent expression of EGFR and IL-13 on airway epithelial cells was observed, and activated EGFR contributed to transdifferentiating of ciliated to goblet cells [23]. Selective inhibition of EGFR inhibited goblet cell metaplasia, synergistically with IL-13 blockade [23]. IL-13 itself does not stimulate EGFR expression or TGF-α, and it is an independent factor for mucus production in transcriptome analysis. However, EGFR and IL-13 shares an inhibitory effect on transcription factor of FOXA2 which regulates mucus production [39]. A recent study using conditional transgenic mice expressing the *IL13* transgene showed that mice expressing IL-13 exhibited airway hyperresponsiveness and neutrophilic airway inflammation which are inhibited by EGF inhibitor treatment [28]. Eosinophilic airway inflammation induced by expressing-*IL13* gene was not improved by treating with an EGFR inhibitor alone, but the combination therapy of EGFR inhibitor and dexamethasone suppressed airway eosinophilia and neutrophilic inflammation. Along with transcriptomic analysis among cohorts, including severe asthma patients, the EGFR pathway makes considerable contributions to mucus metaplasia among asthmatics, as well as the IL-13 pathway [28].

#### 3.1.4. EGFR and Airway Remodeling

EGFR is involved in airway remodeling. An ex vivo study using bronchial specimens from childhood asthma showed that airway EGFR expression increased more in severe childhood asthmatics than control or moderate asthmatics [24]. EGFR expression was positively correlated with airway remodeling assessed by lamina reticularis thickening and collagen III deposition. Interestingly, cell proliferation evaluated by Ki67-positive cells in airway epithelium was not increased in severe childhood asthmatics when compared to the control subjects, which may indicate that EGFR might contribute to airway remodeling phenotype of asthma by stimulating collagen deposition in the airways and not by the proliferative effect of typical EGFR signaling.

#### 3.1.5. EGFR and Airway Eosinophilic Inflammation

Eosinophils play a critical role in chronic airway inflammation and compose a severe asthma phenotype shown as eosinophilic asthma [40]. EGFR also involves interaction with eosinophils, which are important for eosinophilic airway inflammation observed in a severe asthma phenotype. An in vitro study using co-culture with human airway epithelial cell line (NCI-H292) and human blood eosinophils, which were stimulated by IL-3 plus granulocyte-macrophage colony-stimulating factor (GM-CSF) or IL-3 plus IL-5 showed that co-culture with the airway epithelial cells and the eosinophils induced MUC5AC expression, EGFR phosphorylation, and TGF-α production as an EGFR ligand [41]. Treatment using an EGFR inhibitor (AG1478 or BIBX1522) prevented this MUC5AC induction. Treatment using anti TGF-α antibodies also inhibited MUC5AC production. These results indicate that activated eosinophils involve with mucus production through EGFR activation by its ligands including TGF-α. Another in vitro study using the co-culture system with human airway epithelial cells and eosinophilic cell line EoL-1 resulted in increased MUC5AC, platelet derived growth factor AB (PDGF-AB), vascular endothelial growth factor (VEGF), TGF-β1, and IL-8 in culture supernatants [42]. EGFR inhibitor (AG1478) treatment decreased the expression of MUC5AC, PDGF-AB, VEGF, and IL-8. In asthmatic airways, type-2 inflammation induced by Th2 cells plays central roles in allergic airway inflammatory response with Th2 cytokine including IL-4, IL-5, and IL-13. EGFR is also involved with type-2 inflammation. Thymus and Activation-Regulated Chemokine (TARC) is a Th2 cytokine, which drives recruitment of CCR4^+^ Th2 cells to the airways and induced allergic immune responses. Airway epithelial cells stimulated by HDM showed increased TARC expression which was enhanced by IL-4 and TGF-β treatment [17]. Activation of EGFR was also observed by HDM stimulation and selective inhibition of EGFR or a disintegrin and metalloproteinase (ADAM) resulted in downregulation of TARC. These results suggested that EGFR and EGF shedding by ADAM would contribute to type-2 inflammation caused by airway epithelial damaging-external stimuli. Besides airway epithelial cells, eosinophils also express EGFR and play important roles in airway inflammation and airway remodeling. Amphiregulin produced by memory T cells activates EGFR on eosinophils, and the activated eosinophils expressed osteopontin which is involved with fibrotic chronic allergic responses in subjects with allergic airway inflammation [43]. In some cases of severe asthma, neutrophils were also involved with airway inflammation together with IL-8 production [44,45]. In biopsy specimens from severe asthmatics, EGFR expression and neutrophils increased when compared to mild asthma, and EGFR expression was positively correlated with IL-8 expression, suggesting that EGFR contributes to sustained neutrophilic airway inflammation [46]. EGFR may contribute to airway inflammation synergistically in both eosinophilic and neutrophilic pathway. Further studies will be needed to elucidate the clinical implications of EGFR expression in patients with the severe asthma phenotype.

#### 3.1.6. Inhibition of EGFR Pathway Ameliorates Airway Inflammation

Extensive studies have been conducted to elucidate the effect of EGFR inhibitors against airway inflammation, and the downstream pathway of EGFR also has been considered to be treatment targets. Activated EGFR transduces signals through several downstream pathways including extracellular signal-regulated kinase (ERK), PI3-Akt, and Janus kinase-signal transducers and activators of transcription (JAK-STAT) pathways. EGFR is also activated by transactivation with other signals such as Src kinase. A study using an OVA-induced asthma mice model showed that the treatment with EGFR or Src inhibitor improved allergic airway inflammation represented as inflammatory cells in broncho alveolar lavage fluid (BALF), peribronchial inflammation, airway remodeling, and AHR in the same degree [19]. Treatment of inhibitors targeted to EGFR-downstream pathway including ERK1/2, PI3Kδ/Akt, or nuclear factor (NF)-κB also showed an improvement in airway inflammation, but to a lesser extent than EGFR or Src inhibitor. In vitro analysis using airway epithelial cell line, HDM induced inflammatory cytokines of IL6 and IL-8 in a dose-dependent manner, and their expression—along with activation of PI3K-Akt and STAT3—was inhibited by the treatment of with erlotinib and osimertinib [47]. These results indicate that EGFR inhibition might be a more efficient target to control aberrant allergic responses from external pathogens than downstream inhibition of the EGFR pathway. An in vitro inhibitor experiment study using EGFR inhibitors of erlotinib and AG1478 showed that HDM or IL-17 induced granulocyte-macrophage colony-stimulating factor (GM-CSF) production was significantly inhibited by treatments with EGFR inhibitors [20]. Inhibitors of p38 MAPK or tumor necrosis factor alpha (TNFα) converting enzyme (TACE) also inhibited GM-CSF production by HDM stimulation. In in vivo mice experiments carried out in the same study, HDM-induced airway inflammation, airway hyperreactivity, and airway remodeling was improved by the treatment with EGFR inhibitor. The authors suggested that HDM-induced innate immunity activates EGFR-GM-CSF axis through cleavage of EGFR ligands such as amphiregulin on airway epithelial cells. The EGFR-GM-CSF axis might be a potential therapeutic target for allergic asthma. EGFR is also activated by oxidation. Epithelial NADPH oxidase dual oxidase 1 (DUOX1) is a key protein for controlling oxidative stress in airway epithelial cells. In HDM-induced allergic asthma mouse models, oxidative EGFR activation was diminished by treatment with DUOX1 inhibitor with improvement of airway inflammation including neutrophilic airway inflammation and Th2 cytokine production and mucus metaplasia [48]. These results indicate that inhibition of EGFR may be benefit for treatment of chronic airway inflammation observed in asthmatic airways.

#### 3.1.7. EGFR: A Potential Therapeutic Target for Chronic Asthma

Together with the findings of this section, aberrant EGFR activation involves with abnormal airway epithelial proliferation which results in obstructive airflow limitation accompanied by airway remodeling and hypermucus secretion [49]. Excessive EGFR activation may be involved with chronic airway inflammation, which causes chronic asthma. Specific inhibition of EGFR activation may be a potential therapeutic target for chronic airway inflammatory disease including asthma. Indeed, EGFR inhibitors have already been available as cancer treatments with cautions of its occasionally serious adverse effects including interstitial pneumonia and acute lung damage. Local administration of EGFR inhibitor using inhalation devices may be a treatment option to avoid systemic adverse events [26].

### 3.2. ErbB2

Erb2 is the second member of the EGFR family with transmembrane tyrosine kinase receptors. Differently to other EGFR members, ErbB2 is considered to lack its specific extracellular ligand, and to activate ErbB2, autophosphorylation of ErbB2 by dimerizing with other members of EGFR family is needed. Exceptionally, mucin 4 (MUC4) protein acts as an intramembrane ligand for ErbB2. In a study using colon adenocarcinoma cell line (CACO-2), complex forming of ErbB2 and MUC4 phosphorylates ErbB2 on tyrosine 1139 and 1248, and activated the downstream signaling pathway of p38 and Akt [50]. ErbB2 is essential for mammalian development, as a study shows that ErbB2 knockout in mice resulted in embryonic death caused by defects in cardiac and neuronal development [51]. 

The EGFR family coordinates cellular processes upon airway epithelial damage to repair epithelium integrity in the airways. ErbB2 also plays important roles in airway epithelial wound recovery and accumulated evidence shows that ErbB2 involves with airway epithelial cell differentiation as well as cell proliferation and growth, which is indispensable in airway epithelial wound repair. ErbB2 activation induced by its phosphorylation is a key signal for appropriate restoration of airway epithelial integrity after wounding. Vermeer et al. reported that segregation of the ligand and the receptor from the apical to the basolateral side of airway epithelial cells controlled ErbB2 phosphorylation using the airway epithelial cells wounding model [33]. In this study, heregulin was used as a ligand of ErbB3 and ErbB4, which are counter partners of ErbB2. Heregulin and ErbB2 were expressed on apical and basolateral sides of airway epithelial cells, respectively. Besides this, the addition of apical heregulin did not induce ErbB2 phosphorylation. Cell permeabilization allows heregulin to activate ErbB2 phosphorylation expressed on basolateral side of airway epithelial cells. Additionally, ErbB2 phosphorylation increased at the leading edge of wounding, and the treatment with neutralizing heregulin-α antibodies or ErbB2 inhibitors of herceptin decreased wound closure after mechanical wounding. This upregulated ErbB2 expression along the wound leading edge was also observed in other epithelial cell types including corneal epithelial cells [52]. ErbB2 also involved airway epithelial cell differentiation. Primary airway epithelial cells cultured using the air-liquid interface (ALI) system were treated with ErbB2 specific inhibitors of trastuzumab showed decreased ciliated epithelial cells and increased metaplastic cells [34]. Heregulin treatment recovered epithelial dysplasia as shown in increased epithelial cell heights and decreased the number of metaplastic and non-ciliated columnar cells. Co-culture with primary lung fibroblasts, which expressed EGFR ligands including TGF-α, heparin-binding EGF, amphiregulin and heregulin resulted in inducing differentiated airway epithelial cells, which was comparable to heregulin treatment. These results suggest that ErbB2-mesenchymal signaling is a key pathway maintaining airway epithelial differentiation, especially with respect to the ciliated cell population, which is indispensable for removing external stimuli in the airways.

In relation to asthma pathogenesis, several reports described the results that ErbB2 expression is impaired in asthmatic airway epithelium. Dysregulated wound repair in airway epithelial cell culture from asthmatic children was reported in relation to decreased fibronectin expression which is a major extracellular matrix component [11]. In this report, transcriptome analysis using microarray of freshly harvested airway epithelial cells was investigated and ErbB2 was listed in the gene set of significantly decreased genes in asthmatic airway epithelial cells compared to healthy airway epithelial cells. In other studies using airway epithelial cells freshly obtained by bronchoscopic airway brushing among control and adult asthmatics from the Severe Asthma Research Program (SARP), microarray analysis was performed to characterize phenotypes of severe asthma [31]. Using the *k*-means approach to cluster participant samples according to gene expressions, five asthma clusters were identified. A cluster of moderate or severe asthmatics with prominent airway inflammation and lower lung function (named “SC2” in this study) had a differentially downregulated gene expression of *ERBB2* as well as genes of innate and adaptive immunity. A subsequent analysis using the weighted gene co-expression network analysis revealed that a gene module expression linked to epithelial growth and repair including *ERBB2* significantly decreased in severe asthmatics [32]. The expression of this gene module was negatively associated with asthma severity, BMI, and exhaled nitric oxide, and positively associated with FEV_1_% predicted and Asthma Quality of Life Questionnaire (AQLQ) scores. The downregulation of *ERBB2* in severe asthmatic airway epithelial cells was validated by a following study showing dysregulated ErbB2 phosphorylation in freshly obtained airway epithelial cells from asthmatic patients compared to healthy control group patients [10]. Functional analysis using the scratch-wound healing model in ALI culture of primary airway epithelial cells was also conducted in this study. Asthmatic airway epithelial cells exhibited impaired wound healing shown as “slowed” wound closure assessed by 7 h after mechanical wounding on ALI culture surface compared to control airway epithelial cells. Cell proliferation assessed by cyclin D1 (CCND1) protein expression and [^3^H] thymidine incorporation was decreased after wounding in asthmatic airway epithelial cells when compared to the control group. Gene expression of *CCND1* and *ERBB2* was positively correlated in wounded airway epithelial cells. Immunocytological staining of phosphorylated ErbB2 showed that phosphorylation of ErbB2 at the leading edge of wounding was downregulated in asthmatic airway epithelial cells compared to control. Applying a specific inhibitor of ErbB2 phosphorylation (mubritinib) to ALI culture induced delayed wound repair in this scratch-wound model [10].

The precise mechanism of dysregulated ErbB2 expression in asthmatic airway epithelial cells is not well known, but the *ERBB2* gene locus of 17q12-21 (between 35.0 and 35.5 Mb) which is known as the asthma susceptibly gene locus including *ORMDL3*/*GSDMB*, *IKZF3* is a possible explanation for impaired expression among asthmatic subjects [53,54,55,56]. Epigenetic factors observed that DNA methylation around this locus could also be involved with dysregulated *ERBB2* expression among asthmatic population [57]. As a key of epigenetical change in asthmatic airways, airway epithelial cells from patients with prominent airway inflammation represented as higher exhaled nitric oxide in vivo showed worse wound healing using ALI scratch wound assay in vitro [10]. This suggests that sustained airway inflammation epigenetically altered airway epithelial cells to promote lower ErbB2 activation. Additionally, absence of the EGFR family of ligands might also result in delayed wound healing of airway epithelial cells by loss of ErbB2 oligomerization with another EGFR counter partner. RNA-sequencing analysis using sputum samples from HDM-sensitized atopic asthma patients with or without wheezing was performed [58]. In asthmatic subjects with wheezing, two distinct gene expression networks were showed comprising of type-2 inflammation and epithelial associated genes. Among those genes, *EGFR*, *ERBB2,* and *IL-13* had been linked as gene “hubs”, as well as epithelial mucociliary genes including E-cadherin (*CDH1*). Additionally, protein-level validation analysis using primary airway epithelial cells from asthmatic and control subjects was also performed in this study. EGFR and cadherin-related family member 3 (CDHR3) protein expression was increased in airway epithelial cells from atopic asthma subjects compared to control subjects. Although this study was conducted by sputum-derived cell basis, which was not controlled by cell type, especially in epithelial cell rich-asthmatic airways, natural aeroallergen exposure including HDM to asthmatic airways induced disease specific gene expression, which linked type-2 inflammation and airway epithelial gene network.

As the downstream pathway of ErbB2, Janus kinases (JAK) and signal transducer and activator of transcription proteins (STAT) are involved with airway epithelial cell proliferation and differentiation. As a ligand of the EGFR family, neuregulin-1 (NRG-1) induces cell proliferation signals with a high affinity to heterodimer of EGFRs including ErbB2 and ErbB3. In mid-trimester human fetal lung tissue, ErbB2 and ErbB3, but not ErbB4, was expressed on developing lung epithelium [59]. In ex vivo cell culture of airway epithelial cells, NRG-1 stimulation induced EGFR activation as well as cell proliferation and differentiation. In other in vitro studies using airway epithelial cells, NRG-1 stimulated phosphorylation of JAK3, STAT3, and STAT5, and these upregulations were dependent on dimerization of ErbB2 and ErbB3 [60]. Inhibition of the JAK-STAT pathway using specific inhibitors of the JAK-STAT pathway resulted in impaired cell proliferation. Among the NRG-growth factor family, NRG-1β1 was identified as a growth factor for goblet cell formation. Using normal human bronchial epithelial cells (NHBEs), NRG-1β1 induced MUC5AC and MUC5B protein expression in time-dependent and dose-dependent manner [61]. The mucin production by NRG-1β1 was involved with ErbB2 and ErbB3 expression but not with ErbB4. Additionally, downstream signaling of p38 mitogen-activated protein kinase (p38MAPK), ERK1/2, and PI3K also involved with MUC5AC and MUC5B expression through ErbB2 and ErbB3 heterodimerization. Appropriate expression of ErbB2 along with EGFR would be needed for a balanced-airway epithelium cell population between ciliated and mucus airway epithelial cells. Further study will be needed to elucidate whether ligand expression impairment or receptor deficiency could affect dysregulated airway epithelial functions as observed in asthmatic airways. From findings in this section, dysregulated ErbB2 expression in airway epithelium could characterize a phenotype of severe asthma in relation to impaired wound recovery processes, which might result from abnormal epithelial differentiation and mucus production and in turns causes aberrant immune response.

### 3.3. ErbB3 and ErbB4

The relevance between ErbB3 and ErbB4 and asthma pathogenesis is an emerging research field. ErbB3 lacks transmembrane tyrosine kinase activity, considered as being a “kinase dead” receptor, and heterodimerization with other EGFR family members is needed to own phosphorylation. The preferred partner of ErbB3 is ErbB2, and the ErbB2-ErbB3 complex makes stronger signals than other EGFR oligomerization. Research shows that ErbB3 contributes to airway epithelial integrity as shown in epithelial permeability [35]. Treatment with a ligand of ErbB3, heregulin, on airway epithelial cell culture increased transepithelial electrical resistance (TER) and decreased fluorescein isothiocyanate (FITC)-labeled dextran permeability. Interestingly, knockdown of ErbB2, which is a counter partner of ErbB3, resulted in impaired epithelial permeability. These results indicate that heterodimerization of ErbB2 and ErbB3 would implicate with maintaining airway epithelial integrity as shown in epithelial permeability. The other study indicates that ErbB3 also is involved with airway mucus secretion. Airway epithelial expression of EGFRs was investigated among non-smokers and current smokers with or without chronic obstructive pulmonary disease (COPD) [36]. In immunohistochemical staining of bronchial biopsy specimens, EGFR, ErbB3, and MUC5AC expression were significantly increased in smokers compared to non-smokers, with no difference being seen as a result of the presence of COPD. MUC5AC expression was positively correlated with ErbB3 expression among all subjects. Interestingly, there was no difference in the expression of ErbB2, which is preferred heterodimerization partner of ErbB2. The results of this study suggest that not only EGFR but also ErbB3 might be involved with airway mucus secretion, especially associated with exposure to external stimuli including smoking. Recently, ErbB3 expression has a potential association with a severe asthma phenotype. The transcriptome analyses among a multi-center study cohort including severe asthmatics has been reported in relation between the severe asthma phenotype and EGFR family expression [28]. A severe asthma phenotype with neutrophilic inflammation showed upregulated *ERBB3* expression as well as for several EGFR ligands including HB-EGF, epiregulin, and EGF as shown in “IL-17 signature.” Further sub-phenotype analysis revealed that the “IL-13” phenotype and the “IL-17” phenotype had upregulated *ERBB3* expression. Activated ErbB3 has multiple binding lesions for PI3K [62], and the PI3K pathway is involved with airway hyperresponsiveness and airway inflammation [63]. ErbB3 might be a potential therapeutic target for patients with the severe asthma phenotype with IL-17 induced neutrophilic inflammation accompanied by airway hyperresponsiveness.

ErbB4 is also involved with lung development [64,65], and knock-out in mice of ErbB4 is embryonically lethal due to cardiac defect like ErbB2. ErbB4 downregulation leads to reduced production of fetal lung surfactant [66], airway hyperreactivity, and lung inflammation [67], similar to the characteristics of bronchopulmonary dysplasia. The relevance between asthma pathogenesis and ErbB4 has not been well described, but some evidence has been reported regarding the relationship between ErbB4 and asthma. In a genetic association study, genetic variants of *ERBB4* were associated with childhood asthma [68,69]. The gene expression analysis using airway epithelial cells among healthy control and asthmatic subjects showed that a gene network including *ERBB4* designated as “ICS” induced by glucocorticoids was upregulated with severe asthmatics, and its expression was positively correlated with ICS use [32]. An investigation using nasal epithelium from patients with nasal polyp showed that ErbB4 expression was localized in ciliated epithelial cells while EGFR was localized in p63^+^ basal cells [70]. In nasal epithelium from nasal polyp patients, ErbB4 expression significantly increased compared to those from healthy subjects. Corticosteroid treatment normalized ErbB4 expression in hyperplastic epithelium. From a finding of co-localization with ErbB4 and Foxj1 in this study, ErbB4 might contribute to cell plasticity and differentiation during wound recovery in airway epithelial cells. From these result about ErbB4, ErbB4 would contribute to asthma pathogenesis in some degree, especially in severe asthmatics along with use of inhaled corticosteroid, but further basic and clinical studied need to be established for the clinical implication of ErbB4 and asthma.

## 4. Future Directions

EGFRs have been extensively researched as a key pathogenic factor and a potential therapeutic target for asthma. EGFRs are indispensable for airway epithelial recovery from damage by external pathogens including ambient inhaled air. Delayed wound healing of airway epithelium allows those external stimuli remaining into the airway epithelium, subsequently inducing excessive migration of inflammatory immune cells, which often causes chronic airway inflammation shown in asthma. On the other hand, a dysregulated wound recovery response would lead to abnormal wound repair as shown in airway remodeling and hypermucus secretion, both of which are also clinical features of chronic asthma. Two directions of relationship between EGFRs and asthma pathogenesis, “excessive reaction to wounding and impaired wound recovery”, would be speculated for considering future asthma therapeutic target.

First, as excessive reaction to wounding in asthmatic airways, most of the current studies are focused on EGFR expression linked to airway remodeling and hypermucus secretion. EGFR expression increases in asthmatic airways and contributes to airway remodeling and goblet cell hyperplasia. Some studies showed that specific EGFR inhibitor decreased goblet cell numbers. From these results, inhibition of EGFR activation may improve aberrant mucus production and airflow limitation as shown in chronic persistent asthma. Second, as impaired wound recovery, some study showed dysregulated ErbB2 expression related to delayed wound healing in asthmatic airways. ErbB2 is involved with airway epithelial cell differentiation as well as wound recovery of airway epithelium damaged by external stimuli. ErbB2 is indispensable for normal wound recovery processes of airway epithelium and downregulated in airway epithelium among severe asthmatics. Restoration of ErbB2 function may contribute to improve the function of airway epithelial healing, which leads to ameliorate airway inflammation shown in severe asthmatics, especially in atopic asthma patients who have persistent airway epithelial damage by allergic external pathogens (Figure 1). To normalize ErbB2 activation, upregulation of ErbB3 as a preferred partner of ErbB2-heterodimerization might be a potential therapeutic target because of absence of efficient ligands for ErbB2. In terms of ErbB4, there is limited evidence available, but some study suggested the relationship with ErbB4 and severe asthma pathogenesis. Further basic and clinical studies will be conducted.

## 5. Conclusions

In this review, basic and clinical relevance between EGFRs and asthma pathogenesis, in terms of airway epithelial integrity, was discussed. As a front barrier of airways, airway epithelial cells play pivotal roles in asthma pathogenesis and are potential therapeutic targets to maintain airway integrity whose disruption allows external airborne pathogens to remain in contact with airway epithelium and leads to aberrant airway inflammation, bronchoconstriction, and airway remodeling. Airway epithelial cells also contribute to coordinate immune responses to eliminate the pathogen from airway epithelium. Further basic and clinical research would be warranted to accumulate evidence with EGFRs and asthmatic airway epithelial cells, which would help the development as a novel therapeutic target for asthma.

## Figures and Tables

**Figure 1 jcm-09-03698-f001:**
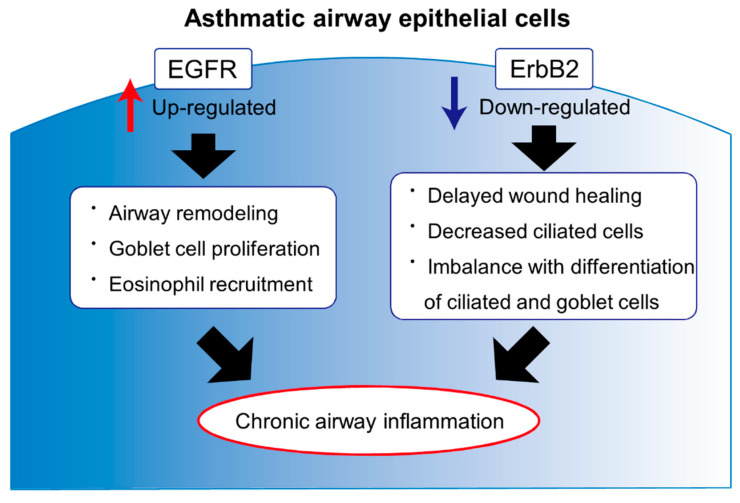
Dysregulated expression of EGFR and ErbB2 in asthmatic airway epithelial cells. Members of the EGFR family play a pivotal role in maintaining airway homeostasis. Asthmatic airway epithelium has impaired integrity, which causes airway remodeling, hypermucus secretion, and delayed wound recovery from external stimuli. In asthmatic airways, expression of EGFR would be increased and involved with airway remodeling, hypermucus secretion, and attraction of eosinophils. Contrarily, expression of ErbB2 would be impaired and involved with dysregulated wound recovery and differentiation to ciliated cells, which would cause remaining airborne pathogens in the airways. Imbalanced expression with EGFR and ErbB2 may be present in asthmatic airways and could be a potential therapeutic target for asthma. Abbreviations: EGFR, epidermal growth factor receptor; ErbB2, erythroblastosis oncogene B2.

**Table 1 jcm-09-03698-t001:** Roles of EGFR family in asthma pathogenesis. Each member of the EGFR family has distinct roles in asthma pathogenesis linking to airway integrity, airway inflammation, mucus secretion, and airway hyperresponsiveness.

	Roles in Asthma Pathogenesis	Reference
EGFR	Increased expression in airway epithelium among severe asthma	[15]
	Increased expression by airway epithelial wounding	[15]
	Eosinophilic airway inflammation	[16,17]
	Airway hyperresponsiveness	[16,18,19,20]
	Airway remodeling	[18,19,20,21,22,23,24]
	Goblet cell proliferation, mucus production	[22,25,26,27,28,29,30]
ErbB2	Downregulated in asthmatic airway epithelium	[10,11,31,32]
	Upregulated with airway epithelial wounding	[10,33]
	Airway epithelial differentiation to ciliated cells	[34]
ErbB3	Epithelial permeability	[35]
	Mucus production	[36]
	Neutrophilic airway inflammation in severe asthma	[28]
ErbB4	Induced by inhaled corticosteroid usage	[32]

Abbreviations: EGFR: epidermal growth factor receptor, ErbB2: erythroblastosis oncogene B2, ErbB3: erythroblastosis oncogene B3, ErbB4: erythroblastosis oncogene B4.

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
