# Peer review of "Airway Epithelial Dysfunction in Asthma: Relevant to Epidermal Growth Factor Receptors and Airway Epithelial Cells"

_jcm, 2020, doi:10.3390/jcm9113698_

Round 1

Reviewer 1 Report

 Authors summarize the relation between EGFR and asthma , in terms of airway remodeling, mucus secretion ,  airway inflammation in an interesting review, focused on the roles of the EGFR family in asthmatic airway epithelium to elucidate the pathogenesis of airway epithelial dysfunction in asthma.

I suggest Authors to mention also growth factors such as hepatocyte growth factor (HGF) and keratinocyte growth factor, (KGF) which play a role in the process of repair of airway epithelium. Tillie-Leblond and Coll ( (ERJ 2007)  demonstrated in chronic asthma that the treatment by KGF before the  OVA nebulisation was associated with a decrease of the leak of epithelial and endothelial markers, suggesting that such a therapeutic strategy, designed to restore the epithelial and endothelial barrier integrity, may have a great potential to limit the consequences for lung functions of increased permeability in severe asthma. Human hepatocyte growth factor (hHGF) is the most potent mitogen identified for alveolar type II cells, and may have other important functions in the repair of the alveolar and small airway epithelium. Okada M (Pediatric Res 2004) showed that  Hepatocyte Growth Factor Protects Small Airway Epithelial Cells from Apoptosis Induced by Tumor Necrosis Factor-α or Oxidative Stress.

The review is interesting and well written.

Author Response

Responses to the comments of reviewer #1:

  1. Authors summarize the relation between EGFR and asthma, in terms of airway remodeling, mucus secretion, airway inflammation in an interesting review, focused on the roles of the EGFR family in asthmatic airway epithelium to elucidate the pathogenesis of airway epithelial dysfunction in asthma.

Response: We are grateful to the reviewer for this favorable comment. As noted by the reviewer, the strength of this review was to elucidate pathogenesis of asthma in relation between airway epithelial dysfunction and epidermal growth factor receptors.

  1. I suggest Authors to mention also growth factors such as hepatocyte growth factor (HGF) and keratinocyte growth factor, (KGF) which play a role in the process of repair of airway epithelium. Tillie-Leblond and Coll ( (ERJ 2007) demonstrated in chronic asthma that the treatment by KGF before the OVA nebulization was associated with a decrease of the leak of epithelial and endothelial markers, suggesting that such a therapeutic strategy, designed to restore the epithelial and endothelial barrier integrity, may have a great potential to limit the consequences for lung functions of increased permeability in severe asthma.

Response: Thank you for that suggestion. We are in agreement with the reviewer’s comment. The growth factors including HGF and KGF also contribute to maintain airway epithelial integrity. We have added this to the introduction section (page 1, line 39).

  1. Human hepatocyte growth factor (hHGF) is the most potent mitogen identified for alveolar type II cells, and may have other important functions in the repair of the alveolar and small airway epithelium. Okada M (Pediatric Res 2004) showed that Hepatocyte Growth Factor Protects Small Airway Epithelial Cells from Apoptosis Induced by Tumor Necrosis Factor-α or Oxidative Stress. The review is interesting and well written.

Response: Thank you for the suggestion. We have added this to the introduction section as mentioned in the previous response (page 1, line 39).

Reviewer 2 Report

In the submitted manuscript, Hideki et al. analyses the state of art of epidermal growth factor receptors (EGFRr) family and asthma. This is an interesting paper summarising the different role that each EGFR family member play in asthma pathogenesis. However, there are some aspect that make difficult to read and follow the information presented.

It is clear that this review has been written by different authors, it is possible to see that the style followed is different in each section. However, as a main concern is that when you are reading a review you expect that all of the sections follow as only one. From the beginning to the section 3.2 is difficult to read it whereas from 3.2 to end the information presented is clearer and more organized. Therefore, the authors should revise and unify the manuscript style.

In line with this general comment, there are some issues in this part of the review that should be corrected.

Abstract, lines 14 to 17. These first and second sentences are duplicated, because both are making reference to the same meaning.

Introduction, lines 43 and 43. “Epithelial EGFRs as classical however potentially novel therapeutic targets”.  

In the section 3.1, there are several sentences difficult to understand:

  • Line 99: “The ligands of EGFR also increase in asthmatic airways as ligands of EGFR”
  • Line 135: “which was inhibited by EGFR specific inhibitor treatment by AG1478”
  • Lines 180-181: “was inhibited by the treatment of EGFR inhibitor [37].” What inhibitor was used?
  • Use of the “treatment of EGFR inhibitor” (line 189). It is more correct use “treatment with the EGFR inhibitor”. This type of sentence is used often in the text as in line 117, “treatment of erlotinib ameliorated”, it is more correct use again “treatment with erlotinib ameliorated”

In is necessary use linking words helping both, authors and reader, to connect the ideas and sentences not only present the information as a simple string of ideas such as in lines 136 and 137 of the section 3.1. 

Author Response

Responses to the comments of reviewer #2:

  1. In the submitted manuscript, Inoue et al. analyses the state of art of epidermal growth factor receptors (EGFRs) family and asthma. This is an interesting paper summarizing the different role that each EGFR family member play in asthma pathogenesis. However, there are some aspect that make difficult to read and follow the information presented.

Response: We are grateful to the reviewer for this comment. The strength of this review was to elucidate pathogenesis of asthma in relation between airway epithelial dysfunction and epidermal growth factor receptors. We have revised the manuscript as the reviewer pointed out.

  1. It is clear that this review has been written by different authors, it is possible to see that the style followed is different in each section. However, as a main concern is that when you are reading a review you expect that all of the sections follow as only one. From the beginning to the section 3.2 is difficult to read it whereas from 3.2 to end the information presented is clearer and more organized. Therefore, the authors should revise and unify the manuscript style.

Response: Thank you for that suggestion. We have revised the style of the manuscript by adding subsections from 3.1.1 to 3.1.7 and organized the style of manuscript to improve the readability of the main text (page 1-6).

  1. In line with this general comment, there are some issues in this part of the review that should be corrected. Abstract, lines 14 to 17. These first and second sentences are duplicated, because both are making reference to the same meaning.

Response: Thank you for the suggestion. We have corrected this duplication in the abstract (page 1, line 16).

  1. Introduction, lines 43 and 43. “Epithelial EGFRs as classical however potentially novel therapeutic targets”.

Response: Thank you for pointing out this. We have revised this sentence as shown in page 2, line 48.

  1. In the section 3.1, there are several sentences difficult to understand: Line 99: “The ligands of EGFR also increase in asthmatic airways as ligands of EGFR”

Response: Thank you for pointing out this error. We have revised this sentence as shown in page 3, line 114.

  1. Line 135: “which was inhibited by EGFR specific inhibitor treatment by AG1478”

Response: Thank you for this comment. We have revised the manuscript as shown in page 4, line 157.

  1. Lines 180-181: “was inhibited by the treatment of EGFR inhibitor [37].” What inhibitor was used?

Response: Thank you for this comment. The EGFR inhibitors used here are erlotinib and osimertinib. We have revised the manuscript as shown in page 6, line 274.

  1. Use of the “treatment of EGFR inhibitor” (line 189). It is more correct use “treatment with the EGFR inhibitor”. This type of sentence is used often in the text as in line 117, “treatment of erlotinib ameliorated”, it is more correct use again “treatment with erlotinib ameliorated”

Response: Thank you for this suggestion. We have corrected the corresponding sentences to “treatment with” as shown in page 4: line 133, line 135, line 140, line 162, page 6: line 267, line 274, page 7: line 278, line 282, and page 8: line 326.

  1. It is necessary use linking words helping both, authors and reader, to connect the ideas and sentences not only present the information as a simple string of ideas such as in lines 136 and 137 of the section 3.1.

Response: Thank you for the suggestion. We have revised the manuscript to present the ideas clearly as the reviewer pointed out (page 4, Line 154, page 2, line 61, page 3, line 98 and 106, page 6, line 262).
